# A Review on Pharmacy Practice in South Africa—A Higher Education Perspective

**DOI:** 10.3390/pharmacy11010003

**Published:** 2022-12-21

**Authors:** Thokozile Mosiane, Kalaba Nkonde, Gauda Mahlatsi, Nontobeko P. Mncwangi

**Affiliations:** School of Pharmacy, Department of Pharmacy Practice, Sefako Makgatho Health Sciences University, P.O. Box 218, Medunsa, Pretoria 0204, South Africa

**Keywords:** pharmacy education, pharmacy practice, pharmacy ownership, pharmacy workforce, rural

## Abstract

In April 1994, South Africa underwent the most significant change in its recent history with the disbandment of the policy of apartheid and the attendant race-based politics, which affected most aspects of the country and, of relevance to this review, the education, health delivery, and career choices that race groups could pursue. In the past 28 years, the South African government has tried to implement policies in order to advance political and socioeconomic shifts toward a more equitable society. The healthcare sector was an early target for transformation that was aimed at increasing access to services and the expansion of primary healthcare and hospital facilities to previously underserved areas. This paper seeks to discuss these changes in broad terms, but with specific reference to general health care and pharmacy practice in particular. It will look at the changes in the legislative framework and pharmacy education and factors impacting the pharmacy practices in South Africa over the past 28 years. A discussion of the critical issues that have affected the profession in the last three decades will also be delineated, and future prospects for the profession as a whole, in terms of pharmacy practice and perspectives, will be discussed. We review the current aspects of the pharmacy profession in South Africa today and how the education of those future professionals is a major contribution to the pharmaceutical climate.

## 1. An Overview of the Pharmacy Profession

Pharmacists are one of the most easily accessible healthcare providers and are the public’s first point of reference for medical advice. In the past, pharmacists were commonly known as chemists that were mainly concerned with compounding. However, over the years, the role of pharmacists has evolved from compounding to product-based (dispensing) practice, and now more recently to a more patient-centered practice [1]. The role of pharmacists now includes ensuring the rational and cost-effective use of drugs, improving clinical outcomes, and improving healthcare by working closely with other healthcare professionals and government entities. Pharmacists have been integrated into the healthcare systems, such as community pharmacies, primary healthcare centers (clinics), academic institutions, and hospitals [2]. 

In the past, South African private pharmaceutical services were mostly concentrated in urban areas, which were mostly white-dominated. Following the 1994 democratic elections in South Africa, the government has sought to address the inequalities in the healthcare system [3]. The attempt to address these inequalities in post-1994 South Africa led to the government passing the Medicines and Related Substances Amendment Act [4] and the Pharmacy Amendment Act [5]. The passing of these two acts resulted in the opening of pharmacy ownership to laypersons and all other healthcare professionals in order to ensure “comprehensive healthcare”. Furthermore, during the discussions, nurses and doctors would not be permitted to issue drugs unless separate pharmaceutical services were not available [6]. While this transformative action was widely accepted by most, it has driven community and private hospital pharmacies to be more product-for-profit centered. Since ownership is open to everyone, large corporations have taken away the sense of offering a service to the community; promotion and recognition are solely based on speed and efficiency, thus making all of the pharmacy interventions that could be affected redundant. This deviates from the patient-centered direction that the sector of community pharmacy is taking. In certain countries, such as Chad, Senegal, and Cameroon, pharmacies can only be owned by pharmacists. In these countries, there is a collective understanding that pharmacists are an integral part of the whole healthcare system and are offering a vital public service [7]. Recently, a study was carried out by Moodley and Suleman [8] to evaluate the impact of the opening of the ownership of pharmacies to laypersons in South Africa. One of the key findings of the study was that opening the ownership of pharmacies to laypersons did not result in a substantial spike or increase in access to pharmacies by previously disadvantaged and rural areas. The clustering of pharmacies in urban areas still existed. The study showed that the ratio of pharmacies to the population was 1.88 pharmacies to 100,000 [8]. 

The National Department of Health (NDoH) handles the issuing of licenses to people who wish to own pharmacies. Licensing restrictions were introduced in 2003 by the government in order to control the geographical area of new community pharmacies. There should be 500 m distance between other dispensing services, and the density is capped at 2 community pharmacies per 10,000 residents—these restrictions come with their exceptions of course [9]. The views on the concepts of equity in health and equity in access to health systems are both cited by the World Health Organization (WHO) as social factors that might create barriers to accessing healthcare services. The WHO utilizes the Tanahashi model of effective coverage [10] in order to describe four different aspects of accessibility to health services that ultimately determine the desired coverage. These are namely the following: (1) the availability of coverage, (2) the accessibility of coverage, (3) the acceptability of coverage, and (4) contact coverage. Therefore, effective coverage can be defined as the proportion of the population in need of healthcare services who ultimately receive a positive health outcome from the intervention. The WHO uses the number of public and private healthcare facilities per 10,000 residents as a key indicator in this model [11]. Ward and associates conducted a study in order to explore the equity in the geographical distribution of community pharmacies in South Africa. Their results suggested that only two provinces in South Africa met the 1 per 10,000 benchmark [12]. In South Africa, we see that the misdistribution of pharmacies persists, despite the growth of corporate community pharmacies and private hospitals. This suggests that a lot is still required policy-wise in order to improve the equity of access to pharmaceutical services in previously disadvantaged areas.

### Legislative Framework

A raft of laws have been passed since 1994 that directly affect the profession of pharmacy. These include amendments to the Medicines and Related Substances Act 101 of 1965, and the Pharmacy Act of 1974, as well as new laws, such as the National Consumer Protection Act 68 of 2008 and, more recently, the regulations for the registration of complementary medicines promulgated in November 2013 [4]. 

The South African government passed the Medicines and Related Substances Amendment Act and the Pharmacy Amendment Act as part of an attempt to repair the imbalance in access to medicines [4,5]. This action by the South African government directly impacted private-sector pharmacies [13,14,15]. Supermarket chain stores have been granted permission to operate pharmacies and, upon the completion of a dispensing course, doctors and nurses can apply for licenses from the NDoH to dispense medication. In addition to these changes, in 2004 the government introduced the Single Exit Price (SEP) legislation in order to limit the profit margins of medicine [4,16]. In combination, these changes provide a considerable threat to the economic survival of private sector community pharmacies under their current product-centered practice model [3,14,17,18,19,20,21].

In 2002, the Medicines and Allied Substances Control Act 101 of 1965 was amended in order to support the government policy of increasing access to medicines. The amendments were intended to control the production, the importation, the distribution, and the sales of medicines. They further relaxed certain patent restrictions and introduced the generic substitution of medicines. They also established a pricing committee to coordinate and oversee a pricing system for all of the medicines sold in South Africa. These changes were not without controversy, and a litigious period followed from 1997, which was when the amendments were first mooted, to 2005. The main point of contestation was the constitutional validity of the pricing committee and the regulations relating to a transparent pricing system for medicines and scheduled substances.

Currently, there are too many pharmacies in the urban areas for the population that is served by them [8,12], and the proposed new pricing scheme is based on the presupposition that, when it comes into operation, the market will become more saturated and pharmacies that are no longer sustainable will close down. The size of the business of those pharmacies that remain will expand and will be sufficient to enable them to increase their profit margins [22].

The International Pharmaceutical Federation (FIP) notes the urgent need for the provision of pharmaceutical care by pharmacists in primary healthcare centers. It also acknowledges the misdistribution of pharmacists and pharmacies within developing countries; most pharmacists would choose to stay in big cities because of the closeness to amenities. The uneven distribution of pharmacies over the urban/rural divide is a typical marker that characterizes developing countries [23].

## 2. Undergraduate Pharmacy Education

The South African Pharmacy Council (SAPC) is the regulatory body of pharmacies in South Africa. On 25 April 2003, The Regulations Relating to the Ownership and Licensing of Pharmacies was published in Notice No. 553, which removed the responsibility of the issuing of a license from the SAPC to the NDoH [24]. Community pharmacies are now obliged to register with the SAPC, and this registration is renewed annually [9]. Education, like health in South Africa, has undergone many transformations. The publication of the White Paper in 1997 and the National Plan for Education in 2001 are examples of that transformation. These papers emphasized the need for new curricula and teaching methods that are inclusive of a larger and more diverse population [25,26].

The SAPC is responsible for developing the entry-level unit standards for pharmacists and the regulation of undergraduate training. These standards were structured for institutions to develop a Bachelor of Pharmacy (BPharm) curriculum that enables a graduate to work in the industry, the community, hospitals, academia, government, and non-governmental organizations. In 2017, the SAPC published for implementation, the Good Pharmacy Education Standards (GPE)—Higher Education and Training in terms of Section 3.4 of the Pharmacy Act, 53 of 1974 [27]. The GPE clearly states the BPharm exit-level outcomes, which were compiled in line with the policy on the good pharmacy education practice of the FIP, in collaboration with the WHO, which identified the “eight-star pharmacist” [28,29]. According to the prescribed exit-level outcomes, qualifying students are required to carry out the following:Integrate and apply foundational scientific knowledge and principles to the pharmaceutical sciences. This includes, but is not limited to, chemistry, microbiology, biochemistry, mathematics, physics, physiology, pathophysiology, anatomy, and social and behavioral sciences, including biomedical ethics;Apply integrated knowledge of product development and formulation in the compounding, the manufacturing, the distribution, and the dispensing of pharmaceutical products;Compound, manipulate, and prepare medication in compliance with Good Pharmacy Practice (GPP) rules, Good Manufacturing Practice (GMP), and/or Good Clinical Practice (GCP) guidelines where they are applicable;Manage and control the development, the manufacture, the packaging, and the registration of pharmaceutical products in compliance with GMP and GCP. The range of pharmaceutical products must include, but is not limited to, medicines, veterinary products, and biological products;Manage the logistics of the selection, the procurement, the storage, the distribution, and the disposal of pharmaceutical products;Dispense medication and ensure optimal pharmaceutical care for a patient in compliance with GPP and, where applicable, GCP;Apply a pharmaceutical care management approach in order to ensure rational medicine use;Initiate and/or modify therapy, where appropriate, within the scope of practice of a pharmacist, and in accordance with GPP and GCP, where they are applicable;Promote public health;Integrate and apply management principles in the practice of pharmacy;Participate in research [27].

The BPharm program is a 4-year-long degree that is offered at nine accredited institutions in South Africa, namely, the Nelson Mandela University (NMU), North-West University (NWU-Potchefstroom Campus), Rhodes University, Sefako Makgatho Health Sciences University (SMU), Tshwane University of Technology (TUT), University of Limpopo (UL), University of KwaZulu-Natal (UKZN-Westville Campus), University of the Western Cape (UWC), and the University of Witwatersrand (Wits). Six of the nine universities offer pharmacy training in a passive learning environment (objectivists’ view of knowledge), while SMU, TUT, and UL offer an integrated problem-based program (the constructivists’ view of knowledge). After completing the degree, the graduates are compelled to complete a year of internship in any training-approved sector and to complete a pre-registration examination that is compiled by the SAPC. Upon the successful completion of the internship, the graduates can register with the SAPC as pharmacists and complete a year of community service in a government/state hospital [30,31].

As mentioned above, the BPharm exit-level outcomes are meant to produce a generalist graduate who can work in all sectors, but is there congruence in how the BPharm program is offered in the different universities? Although the SAPC has a set guideline for the exit-level outcomes, every school or faculty of pharmacy across the country has a certain approach to the BPharm program, which makes transfers between the different schools or faculties very difficult, or even impossible.

Table 1 compares the different modules that are given in South African universities in the first year and the fourth year of study [32,33,34,35,36,37,38]. Tshwane University (formerly known as Technikon Pretoria) and the University of Limpopo (University of the North) were omitted from the list as both of these universities offer similar modules to SMU—these universities were part of a merger that occurred in 2005, hence the similarities in module content. The majority of these universities offer “science” modules in the first year of study, except for SMU. The pharmacy students often attend physics and chemistry classes with students from other science streams in their first year of study, as the content is similar. In the first year of study, we see how the different schools or faculties of pharmacies have different emphases.

As shown in Table 1, there is a complete shift when we reach the fourth year of study; the modules that are offered in the different universities are almost identical. Only in the fourth year of study do we see some similarities or congruency in how the BPharm program is offered in the different universities. Towards the fourth year of study, all of the students are encouraged to complete their experiential learning in different sectors of pharmacy in order to prepare them for the pharmacy profession. An emphasis is also placed on experiential learning on the fourth level of study. From their first year, the students are required to partake in experiential learning (work-integrated learning) in a registered community or hospital pharmacy.

Figure 1 is a depiction of the number of hours that the students would spend on experiential learning before the 2013 curriculum change. One could ask if this time that is allocated towards experiential learning was enough to equip the graduates with enough knowledge and skills to enter the workforce. The majority of the schools of pharmacy that are depicted in Figure 1 were mainly focused on experiential learning in a hospital and community pharmacy; sectors such as industrial pharmacy and primary health care clinics were not given much attention [39].

As newly trained pharmacists enter the health system annually, it is paramount that the curriculum developers ask themselves if they are equipping the graduates with enough skills to enter the workforce as generalists. Currently, pharmacy training does not involve much business, leadership, or human resources; however, after completing the community service year, a pharmacist is expected to have those skills, knowledge, and competencies. In addition, teaching and learning with technology is not enforced at pharmacy schools, and this puts the existing students at a disadvantage, especially where they have to dispense using the company’s software. Even though this can be taken as on-the-job training, most employers prefer a candidate who knows how to work on their system (e.g., ProPharm, UNISOLV). Therefore, we ought to ask ourselves if we are producing pharmacists who are equipped to work effectively in all pharmacy sectors.

## 3. Emerging Issues

### 3.1. Growing Need for Affordable Medicines

The emergence of HIV and AIDS has exposed the healthcare system in South Africa to its inadequacies. The rollout of anti-retroviral medication since 2009 has put pharmacy at center stage, highlighting some of the shortcomings of the profession. The roll-out, as well as changing the epidemiological patterns and the acceptance of generic products over the past 20 years [40], has made South Africa a lucrative market for the global pharmaceutical industry. The net effect of this has been an increase in submissions for registrations that are received by the Medicine Regulatory Authority (MRA). In 2005, 613 applications were received; this number doubled in 2006 to 1162; and by 2013 it was over 1300. Due to low capacity and logistical challenges, the MRA has failed to cope, resulting in a backlog in processing applications of over 2000 and extended delays in medicine registration with both financial and access implications [41].

The regulatory challenges have highlighted the inadequacy of pharmacists’ training in medicine regulation. Regulatory pharmacists are scarce in South Africa, which is mainly because adequate training is not given in the regulatory sector at the undergraduate level. The current curricula do not seem to be adequately preparing the graduates for this important and relevant area of practice. There is talk of establishing an academy for training regulatory pharmacists in order to address this problem.

A related issue that has emerged over the past 10 years, and with the government’s push to improve access to medicines throughout the republic, is that of supply chain management. Stockouts of essential medicines are a common phenomenon [42], and this is largely due to poor procurement and stock management processes. Again, this seems to point toward weaknesses in the training of pharmacists.

Both of these issues need to be addressed by offering post-graduate and continuing education for the pharmacists who choose these areas of practice. Financial incentives and professional recognition should be used as rewards. While the SAPC is pursuing the registration of specialties, there is a need to re-think the relevance of some of them to South Africa, where the challenges are less at the clinical practice level, but rather at access points (i.e., the regulation and logistician pharmacists are probably equally as important as the clinical pharmacists, as their availability and intervention will have an impact on improved and increased services to a wider population).

### 3.2. Telehealth and Artificial Intelligence Play Growing Roles in Medication Management

The COVID-19 pandemic hastened the use of technology in health (telehealth), increasing the likelihood that health systems will use digital health solutions and robotics in order to optimize medication management. Pharmacists and pharmacy personnel are likely to be charged with leading such efforts to streamline digital health solution integration into the current workflow in order to enhance customers’ experience [43]. In recent years, the prominence within pharmacy has moved away from product-oriented services and is steadily realigning towards the provision of clinical care. With the introduction of services such as Primary Care Drug Therapy Pharmacist (PCDT), vaccinations, COVID-19 testing, and possible pharmacist-initiated management of antiretroviral therapy (PIMART) services, the drive toward a clinical focus in pharmacy is well underway. Pharmacists form part of an under-utilized and highly qualified human resource that can be utilized more efficiently and appropriately in order to benefit the healthcare system and the public [44].

### 3.3. The necessity of the Community Service Year

According to the SAPC, the patient-to-pharmacist ratio in the country is about 30 pharmacists per 100,000 people. As of September 2022, the SAPC indicates that Gauteng has the highest number of pharmacists, at 6354. This includes specialist pharmacists (specializing in either pharmacokinetics or nuclear pharmacy) however, excludes interns and community service pharmacists [45]. The resultant effect means that the rural and the underserved areas of South Africa do not have the required ratio of pharmacists for patients,, which has been an ongoing national issue. Early on in the new democratic South Africa, the concept of community service was introduced for pharmacy graduates in order to bridge the divide in the patient-to-pharmacist ratio. However, in recent years, with the new community service allocation system, a lack of/frozen posts in the public sector, and an increased number of graduates, more completing interns are left without employment and communities are left without pharmacists [46]. The Pharmaceutical Society of SA in 2017 stated that if the government could not afford to finance community service posts, it had no legal or moral basis to enforce community service and it should end the practice. This point is further reinforced by allowing the allocation of community service pharmacists to private-sector community pharmacies due to the lack of state posts. However, the initial objectives are to ensure the improved provision of pharmaceutical services, especially to rural and underserved areas, and to provide young professionals with a chance to develop their skills, and to gain knowledge, reasoning skills, and to find their footing in the pharmaceutical sector and begin to establish their careers, which would be diminished [46]. In a review by Chopra and associates, it was revealed that South Africa has made some progress towards several intercostal millennium development goals for health; however, such progress has been insufficient or even reversed for many health goals, such as those mentioned above [41].

### 3.4. Pharmacy Specialties in South Africa

South African pharmacists can work in several areas, including retail community pharmacy, hospital pharmacy, academic pharmacy, and industrial pharmacy. Most pharmacists work in hospital pharmacies and retail community pharmacies. A smaller proportion makes it to the pharmaceutical industrial pharmacy (as production pharmacists, regulatory affairs pharmacists, quality assurance pharmacists, and responsible pharmacists), as well as academic pharmacy (as lecturers and researchers at various levels). Currently, a specialty that is registrable with the South African Pharmacy Council is clinical pharmacokinetics, although no tangible posts for this specialty exist in state institutions.

There are two registrable categories of pharmacist specialists, namely clinical pharmacokinetics and radio-pharmacist, according to the SAPC 2020 report [47].

Primary Care Drug Therapy pharmacists are growing by the year and are entitled to prescribe medications up to schedule four (prescription-only medicines) for a limited number of diseases on a primary healthcare list. There is also a new addition to the specialties of pharmacists—PIMART, which gives pharmacists an expanded scope of practice. The new and proposed specialties include clinical pharmacy (including pharmacokinetics), pharmaceutical service in public health, and industrial pharmacists. Pharmacy practice is not yet a specialty that is recognized by the SAPC.

## 4. Recommendations

The following recommendations are made based on this review and the supporting articles:Curriculum harmonization throughout all of the BPharm program offerings in order to ensure the same exit-level outcomes for learners across South Africa. A swift move to a 6-year-long professional doctoral program (PharmD) would offer opportunities to incorporate aspects of pharmacy practice such as supply chain management and regulatory affairs in detail before the workplace setting. This would see the role of the pharmacist greatly expand for this generation of pharmacists;Expedite the registration and the legalization of the proposed new specialties and targeted recruitment of pharmacists to pursue postgraduate studies and specialize in them as more subject matters help to build experts in the field;The functions of higher education institutions should be formalized in terms of attracting and choosing students from rural regions, exposing them to rural amenities during their undergraduate training, offering postgraduate chances for professional development during the community service year, and increasing the rural allowance to an attractive monetary figure;Incorporating the concept of inter-professional disciplinary teams in the curricula of all healthcare professions will cement the idealism of the importance of the respective professional roles and responsibilities, as well as advocate the development of a sustainable pharmacy workforce that is relevant to current health needs.

## 5. Conclusions

This review aimed to highlight the current pharmacy practice landscape in South Africa through a higher education lens. It has provided an opportunity to explore the current BPharm curricula and highlight any challenges or needs that may arise in the coming years. It has also shown how inadequate mechanisms to produce, register, supply, and distribute medicines along the supply chain to the public health facilities cause South African medicine provisioning issues, which are not just tied to the medicine budget restrictions but also to inadequate medicine regulation processes and a lack of education thereof. Moreover, while mandatory community service is a good way to draw health professionals to underprivileged and rural regions, it is ineffective at keeping them there in the absence of additional, longer-term human resource reforms.

## Figures and Tables

**Figure 1 pharmacy-11-00003-f001:**
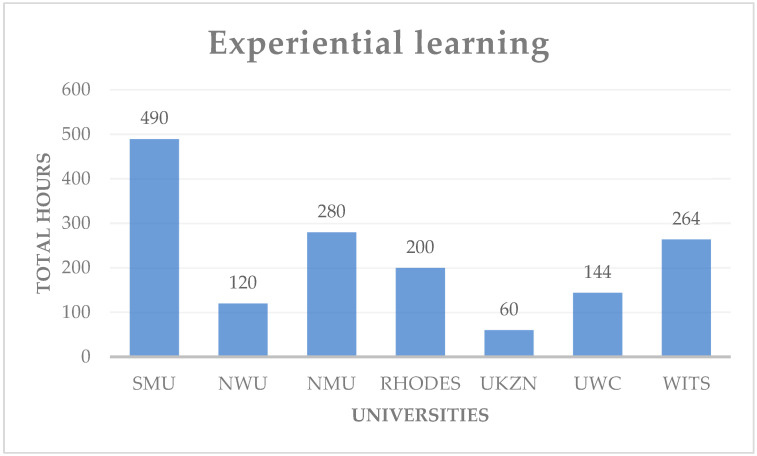
The total number of hours spent on experiential learning over four years in selected universities.

**Table 1 pharmacy-11-00003-t001:** A brief comparison of BPharm curricula in selected South African universities.

Universities	First-Year Modules	Fourth-Year Modules
**NMU**	Pharmaceutical chemistry Physics Pharmacy people and systems Anatomy and physiology Physical pharmacy Computing fundamentals Organic chemistry Biochemistry	Pharmaceutical sciences I Pharmacy and the professional environment Clinical pharmacy Pharmaceutical science II Clinical placement Research project Electives
**NWU**	Academic literacy development Physiology for pharmacy Pharmacy practice I Microbiology for pharmacy Introductory inorganic and physical chemistry Pharmaceutical calculations Clinical pharmacy Introductory organic chemistry	Pharmaceutical chemistry Pharmacology Pharmaceutics Clinical pharmacy Pharmacy practice IV Integrated pharmaceutical care Research project
**RHODES**	Chemistry I Cell biology Introduction to information and communication technology Mathematics for life sciences Anatomy and physiology Biochemistry Chemistry II	Pharmacology I Pharmacotherapy Research project Elective Pharmaceutics Pharmacy Practice Pharmacology II
**SMU**	Introduction to pharmacy From atoms to medicine Biopharmaceutics, pharmacokinetics, and pharmacodynamics Microorganisms, man, and medicines Nutrition and gastroenterology English	Neurological and psychiatric pharmacy Hospital pharmacy practice Specialized pharmacy First aid Hospital-based pharmaceutical care Research project Hospital pharmacy practice-based learning
**UKZN**	Introductory biology for health sciences Mathematics for natural sciences Pharmaceutical chemistry Intro physics for life sciences and agriculture Introduction to pharmacy Health and illness behavior Pharmaceutical chemistry Isizulu language studies	Biopharmaceutics Applied pharmaceutical care Pharmacology Advanced pharmaceutics Natural products and evidence-based medicine Pharmacology Research project
**UWC**	Introduction to Afrikaans/IsiXhosa Chemistry I Human biology Chemistry II Introduction to pharmacy Pharmacology and clinical pharmacy Introductory physics for pharmacists Mathematics	Pharmaceutics Pharmacy Pharmaceutical chemistry Pharmacology and clinical pharmacy Pharmacy practice Electives
**WITS**	Introduction to medical sciences Chemistry I Physics Pharmaceutical practice Health systems sciences Introduction to medical sciences Chemistry II	Pharmaceutics Pharmaceutical chemistry Research project Clinical pharmacy Pharmacy practice Pharmacology

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
