# Peer review of "A Review on Pharmacy Practice in South Africa—A Higher Education Perspective"

_pharmacy, 2022, doi:10.3390/pharmacy11010003_

Round 1

Reviewer 1 Report

Great perspective on the history of South Africa. Very interesting to read.  

A few specific considerations:

- Line 18 , abstract, remove "on" from factors impacting

- Line 31, introduction, add "practice" to patient-centered

- Line 109 - FIP I don't think it was spelled out until line 130. 

- Line 131 - I would switch this to WHO as you have already used this abbreviation. 

- Line 163, I am not sure it is evident that the 4th year is structured in a way to prepare the student to be a professional - just using the title of the courses? I think that would require course objectives. Perhaps the stronger argument is the experiential hours. This is understood to be the culmination and application of skills. 

- As someone outside, perhaps it would be helpful to share what is the standard exit level outcomes. Are there #s of experiential education in place, are they split up by setting or totally up to the institution. You mention few take place in primary care - are there enough places for students to go for those types of experiences?

- Table 1: Looks like it is missing labels across the left handed side. It makes it difficult to know how the authors placed certain courses in certain buckets, or is this just a giant list of the named courses? Some are repeated, which makes it confusing. I am not sure this table helps? I know you want to show how the 1st year is less consistent than the 4th year, but I am not sure this does that or if that point is necessary in the context of the whole. I think just explaining what are the standards set for pharmacy education. 

-Line 207, not sure I understand "in the main"

- Lines 200-203, can you clarify exactly what is inadequate training in medicine regulation? Do you mean the rules or strategies to overcome? 

- Line 231 - this is spelling out an abbreviation that has been used earlier in the text.

- I like the recommendations being summarized. I am not sure I saw supportive evidence or even mention of a PharmD, so recommendation ii. seems out of place. Could it not be placed into the BPharm harmonization? 

- Recommendation iv. is not a recommendation? 

- Recommendation v. ends with a comma. Also, I am not sure evidence was presented that selecting a student from a rural region will ensure they will return. Sometimes it is more about the job market. I would rather see this recommendation be a call for implementing strategies for increased rural offerings or something like that. I think the community service year could be it's own recommendation (to find ways to continue sustainable positions). 

Reviewer 2 Report

This is an excellently written manuscript. I have just a few minor suggested changes:

1. on page 2, line 46, change weren't to were not, contractions are not appropriate in scholarly manuscripts. 

2. on page 3, line 104-106, suggest including reference for excessive numbers of pharmacies.
